# Patients' experience of and participation in a stroke self-management programme, My Life After Stroke (MLAS): a multimethod study

Emily Grace Blatchford  ,[1] Maria Raisa Jessica Aquino,[2] Julie Grant,[1] Vicki Johnson,[3] Ricky Mullis,[1] Lisa Lim,[1] Jonathan Mant[4]

¹Primary Care Unit, University of Cambridge Primary Care Unit, Cambridge, UK
²Population Health Sciences Institute, NIHR Applied Research Collaboration North East and North Cumbria, Gosforth, UK
³Leicester Diabetes Centre, University Hospitals of Leicester NHS Trust, Leicester, UK
⁴General Practice and Primary Care Research Unit, University of Cambridge Primary Care Unit, Cambridge, UK

**Correspondence to**
Emily Grace Blatchford;
egb39@cam.ac.uk

## ABSTRACT

**Objective** A self-management programme, My Life After Stroke (MLAS), was developed to support stroke survivors. This evaluation reports patients' experience.

**Design** Multimethod, involving interviews and questionnaires.

**Setting** 23 general practices in the intervention arm of a cluster randomised controlled trial in East of England and East Midlands, UK.

**Participants** People on the stroke registers of participating general practices were invited to attend an MLAS programme.

**Interventions** MLAS comprises one-to-one and group-based sessions to promote independence, confidence and hope.

**Primary and secondary outcome measures** The primary outcome was uptake of the programme. Participants who declined MLAS were sent a questionnaire to ascertain why. Attendees of four programmes completed evaluation forms. Attendees and non-attendees of MLAS were interviewed. Ad-hoc email conversations with the lead author were reviewed. Thematic analysis was used for qualitative data.

**Results** 141/420 (34%) participants (mean age 71) attended an MLAS programme and 103 (73%) completed 1. 64/228 (28%) participants who declined MLAS gave reasons as: good recovery, ongoing health issues, logistical issues and inappropriate. Nearly all attendees who completed questionnaires felt that process criteria such as talking about their stroke and outcomes such as developing a strong understanding of stroke had been achieved.

**Conclusions** MLAS was a positive experience for participants but many stroke survivors did not feel it was appropriate for them. Participation in self-management programmes after stroke might be improved by offering them sooner after the stroke and providing a range of delivery options beyond group-based, face-to-face learning.

**Trial registration number** NCT03353519, NIH.

## INTRODUCTION
### Background
There is a lack of support for stroke survivors living in the community.[1–3] Stroke survivors

---

### STRENGTHS AND LIMITATIONS OF THIS STUDY

⇒ The self-management programme used in this study was developed using extensive patient and public involvement and has been demonstrated to be feasible to deliver.
⇒ The study population is people on general practice stroke registers rather than those recruited from specialist stroke services, so better reflects the experience of a self-management programme of longer-term community-dwelling stroke survivors.
⇒ The study does not provide data on effectiveness of this self-management programme, but on uptake and patients' experience.
⇒ The multimethods approach allowed us to understand the reasons that people do not take part in these programmes.

---

report a sense of abandonment and a need for social, psychological and educational support.[4–7] Psychological issues are associated with higher rates of mortality, long-term disability, hospital readmissions, suicide and higher utilisation of outpatient services.[8–10] The need for development and evaluation of different models of care to address long-term needs of stroke survivors in primary care is well recognised,[11] yet such models remain scarce.[12] Adjusting to life after stroke continues after formal rehabilitation has finished.[13 14] The evidence base for the role of self-management programmes is growing[15]: such approaches have been successfully applied to long-term conditions such as diabetes[16 17] and can reduce pressure on social and healthcare services[18] by allowing participants to take an active role in their own condition, making informed choices while adopting behaviour and psychological changes.[15 19–21]

Self-management programmes have been used to support stroke survivors during their aftercare. Programmes such as the New

Start[22][23] intervention, the Australia-based Stroke Self-Management Programme (SSMP),[24] Bridges[25] and the Canadian Moving On after STroke (MOST)[26] include processes such as increasing contact time, identifying needs, exploring social networking, problem-solving, goal setting, forming action plans, providing stroke-specific information, using trained healthcare professionals and increasing exercise. MOST-Telehealth Remote[27] took self-management virtually using video conferencing to remotely connect facilitators with participants further afield.

As part of a new primary care-based model of care, Improving Primary Care After Stroke (IPCAS),[28] a novel self-management programme, My Life After Stroke (MLAS) was developed using the Medical Research Council framework and extensive patient and public involvement (PPI), to target the long-term needs of stroke survivors.[29] Areas covered include social well-being, community reintegration, emotional support, physical needs, activities of daily living and prevention of future strokes, informed by self-efficacy and cognitive–behavioural therapy (CBT). This intervention is one of few that focus on stroke survivor's self-reported needs. MLAS was subsequently found to be feasible to deliver[30] and so was included as one of the components of IPCAS that was evaluated in a randomised controlled trial.[28]

## Objective

While it was not possible to directly evaluate any of the individual components within the IPCAS model of care using the randomised design, we wanted to better understand the potential role of self-management programmes such as MLAS. This paper therefore focuses on patients' experience of and participation in MLAS as part of a larger process evaluation.[31]

## METHODS
## MLAS

MLAS programmes (n=24) were held between September 2018 and January 2020 across 6 counties in England: Norfolk (n=8), Suffolk (n=4), Northamptonshire (n=4), Cambridgeshire (n=5), Leicestershire (n=2) and Essex (n=1). Programmes lasted 9 weeks with over 11 hours' contact time.[29] There is an individual appointment with a facilitator at the start of the programme, and a second one at the end. In between individual appointments, there are four group sessions (2.5 hours each) covering different topics. The completion criterion states participants must attend both individual appointments, group session 1 and a minimum of two other group sessions. Individual appointments (face to face or telephone) consist of a discussion to build rapport and explore the impact of their stroke before and after MLAS. Group sessions were interactive, consisted of social support and signposting to local services and topics including managing physical and emotional health, warning signs and risks of stroke. All sessions aimed to build confidence,

independence and hope through problem-solving, CBT-based approaches and opportunities to share personal experiences. The same two facilitators ran all six sessions of a programme (where possible) and were trained and mentored in behaviours such as being non-judgemental, not giving specific advice, effective communication and active listening skills and how to use the MLAS facilitator curriculum and resources to deliver the intervention as intended. Participants were provided with refreshments, programme handbooks and each session began with a recap of previous sessions.

## Recruitment

Participants were in the intervention arm of the IPCAS cluster randomised trial.[28] People were eligible to take part in IPCAS if they were on the stroke register of a participating general practice, were 18 years or older and able to provide written informed consent, with or without the help of a carer. They were invited to attend a programme while at a stroke review led by a healthcare professional at their General Practice surgery. Invitees were given the study team's contact details and the lead author (EGB) contacted all participants 2 weeks post invitation by telephone or email. Those expressing interest were booked onto a programme in an accessible (eg, parking, wheelchair-friendly and accessible facilities) community venue near their general practice surgery. Prearranged and funded return transport was available. Carers and/or significant others were encouraged to attend with participants.

## Facilitators

Facilitators (n=23) were trained by the intervention developers during 2018/2019.[29] Facilitators' backgrounds included nursing, psychology and those with specific interest in stroke.

## Methods of data collection

Qualitative and quantitative data were collected in order to answer the questions below. The qualitative data included a subset of interviews that had been carried out for the wider process evaluation.[31] This included eight participants who completed MLAS, four who started, but did not complete and five who did not take part (two of whom had expressed interest).

### Why did participants decline MLAS?

In addition to interviews with five non-participants, a questionnaire was sent 6 months post randomisation to all participants which included a question asking if they had not attended MLAS, why that was (see online supplemental appendix A). In some areas, there were delays with set-up of MLAS which meant that participants' received the questionnaire before their invitation to attend the programme and so were unable to provide a response to this question.

### What did participants experience when attending MLAS?

All attendees from a purposefully selected sample of programmes (n=4) received an evaluation form

to complete at the end of each MLAS session, which explored what happened during the session and what they took away from it. Programme selection criteria included facilitators' leading programmes and geographical location. We also aimed to include participants who were not participating in other elements of the wider process evaluation (eg, interviews) to offer individuals opportunities to share their experiences, and at the same time, minimise participants' burden. Responses were on a Likert scale and included a comments box (see online supplemental appendix B).

### What are participant views on MLAS?

Participants in the process evaluation were invited to take part in semistructured telephone interviews. Potential participants were purposively selected to ensure that people who completed a programme, started a programme but did not complete it, declined to take part or agreed to take part but did not attend were all included. In order to facilitate engagement of people with aphasia or cognitive impairment, participants could select how they were interviewed (with or without carer; by telephone or face to face or by email). Interviews were performed by JG using a bespoke topic guide to aid open-ended discussion suitable for differing levels of cognitive function. The topic guide was piloted with a stroke survivor to ensure suitability of questions. Topics covered experience and participation (see online supplemental appendix C), informed by the Behaviour Change Consortium framework.[32]

### What feedback was given after MLAS was delivered?

Informal general feedback was given to the lead author (EGB) from both participants and facilitators via email post attendance.

### Analysis

We applied the six-phase approach of Braun and Clarke[33] to the data as follows: (1) data familiarisation where transcripts and questionnaires with qualitative data were read and reread; (2) generation of initial codes, where two coders independently generated codes on a subset of transcripts, then refined these and applied the codes to the rest of the data; (3) identified themes, that is, searching for patterns within the coded data; (4) reviewing the themes to refine each theme's meaning and to determine whether some themes lacked coherence/depth and needed discarding; (5) defining and naming themes which involved finalising the set of themes, discussing any disagreements between coders with a third member of the research team (MRJA); and finally (6) writing up the report which was led by the first author (EGB). Patients' receipt questionnaires were analysed for process and outcome data and simple descriptive statistics were used for the quantitative data. A two sample t statistic was used to compare if there was a difference between those who attended and did not attend, in relation to the distance from individuals' home to the programme venue and in relation to time since stroke.

### Patient and public involvement

We actively engaged PPI in the development of the Managing Life After Stroke self-management programme through lay membership of our Programme Steering Committee and our Intervention Development Group, a lay representative coapplicant, active engagement with The Stroke Association and in several meetings with stroke survivor and carer support groups in the community.

Consultation meetings took place with stroke survivors in April 2016 (2 in Leicester, 1 in Cambridge—20 people per group) to seek input on content of the self-management intervention and its format.

The topic guides for our interviews were pilot tested with PPI members via email, telephone calls and a mock interview.

### RESULTS

Overall, 522 stroke survivors were randomised to the intervention arm. Overall, 59 did not attend the stroke review, 42 withdrew from the trial and 2 died. As a result, 420 individuals were invited to attend an MLAS programme. Overall, 192 (46%) participants who were invited to MLAS expressed interest in attending (figure 1). Of those expressing interest, 141 (73%) attended and 103 of those attending (73%) completed the programme. Overall,

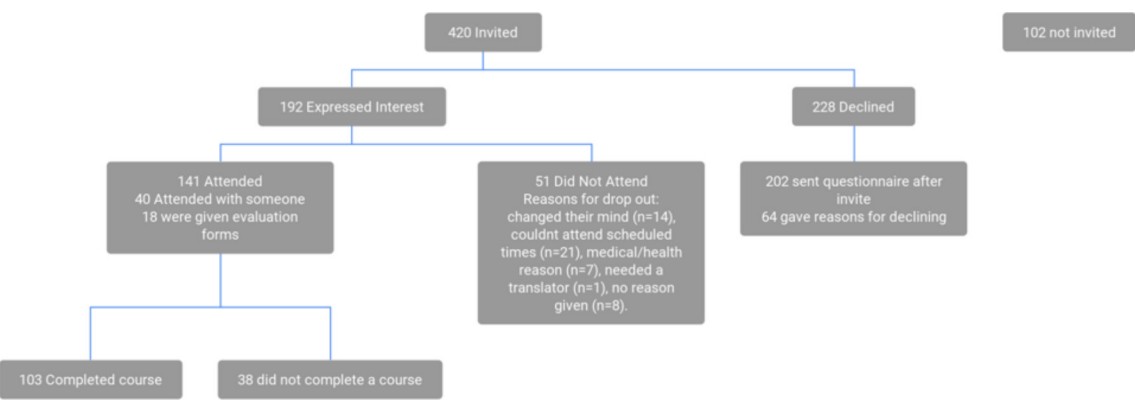

**Figure 1** Study flow chart for those invited, attended and declined My Life After Stroke.

**Table 1** Characteristics of stroke survivors by whether or not they attended the MLAS programme

| | Attended (n=141) | Expressed interest but did not attend (n=51) | Did not wish to attend (n=228) |
|---|---|---|---|
| Gender | | | |
| Male | 62.4% (n=88) | 62.7% (n=32) | 66.7% (n=152) |
| Female | 37.6% (n=53) | 37.3% (n=19) | 33.3% (n=76) |
| Ethnicity | | | |
| White | 97.9% (n=138) | 100.0% (n=51) | 96.5% (n=220) |
| Other | 2.1% (n=3) | 0.0% (n=0) | 3.5% (n=8) |
| Age | | | |
| Mean years at baseline, (SD) | 71.2 (10.4) | 70.9 (12.3) | 69.7 (11.7) |
| Time since stroke | | | |
| Mean years since stroke to study entry (SD) | 7.7 (9.0) | 6.6 (7.3) | 8.2 (8.5) |
| Distance to MLAS venue | | | |
| Mean miles from residence to MLAS venue* (SD) | 2.3 (3.6) | 3.5 (5.1) | 2.8 (4.0) |

*Excludes patients from one practice where 13 patients were invited, but the programme was never run.
MLAS, My Life After Stroke.

24 separate MLAS programmes were held. The median attendance at a group session was 4.5 (range 2–8).

Participants' characteristics are shown in table 1. A total of 94% (133/141) participants attended a programme more than a year after their stroke. There were no differences between those who attended and did not attend. A total of 70% (99/141) participants attended a programme within 2 miles from their residence. The mean distance from the programme venue to the stroke survivor's home and time since stroke were not significantly different between those who did and did not attend the programme (p value 0.26 and 0.50, respectively).

### Why did participants decline MLAS?
Sixty-four participants gave multiple reasons for declining MLAS, which were grouped into four main themes: good recovery, ongoing health issues, logistical issues and not appropriate (figure 2).

### Good recovery
Overall, 28 people felt they made a good or full recovery, experienced no lasting side effects and their stroke was a long time ago; therefore no longer needed further support.

### Ongoing health issues
Overall, 13 people had ongoing health or medical issues such as age-related concerns, cancer treatments, short-term preplanned hospital visits or minor illnesses.

### Logistical issues
Nine people were unable to attend due to other commitments such as carer duties, the programme being too time-consuming, inconvenient days/times and travel issues.

### Not appropriate
Overall, 22 people did not feel the programme was appropriate for them, including disliking group sessions, lacking confidence, uncertainty about the benefits of attending and concerns about the programme being too distressing.

The most common reason given by people who had indicated they would like to attend but subsequently did not was that it was being held at an inconvenient time. Views expressed on self-management programmes in general by non-attenders were positive; they thought they

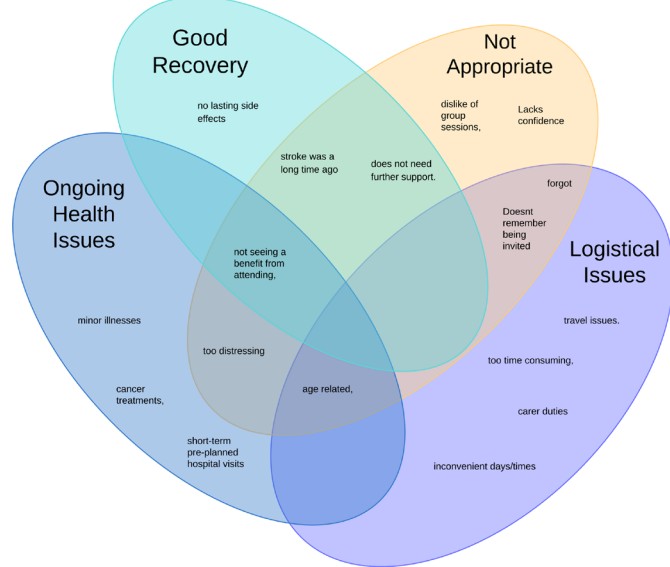

**Figure 2** Venn diagram of themes for declining the self-management programme, My Life After Stroke.

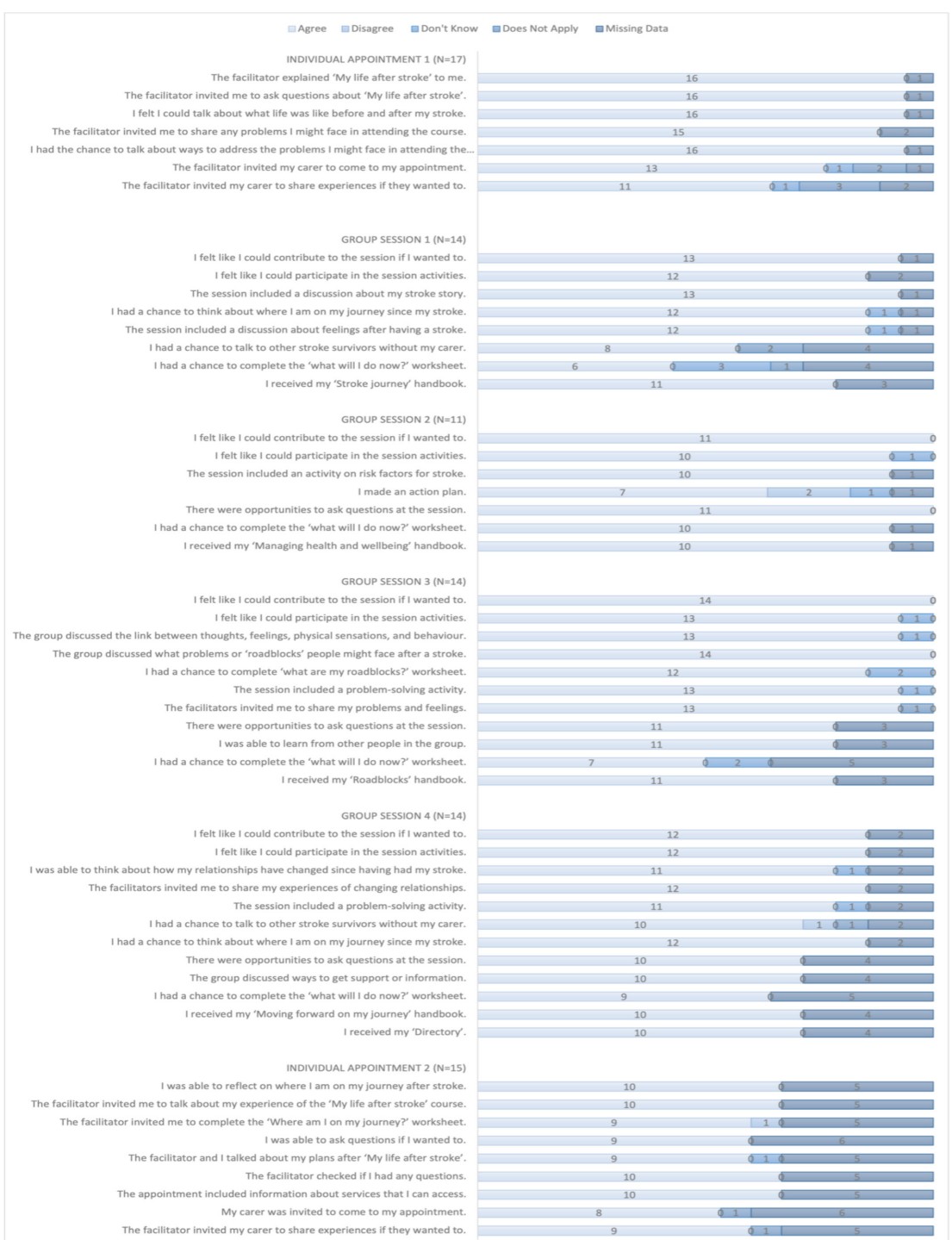

**Figure 3** Patients' evaluation forms, process questions. N=number of participants attending the session.

would be useful, tailored to individual needs, good for social aspects and learning.

The interviews with participants did not attend MLAS gave reasons matching those in figure 2, with additional ones including: the programme having no connection to their stroke, previously attending similar programmes, being in full-time employment and moving away.

### What did participants experience when attending MLAS?

Figures 3 and 4 show the extent to which participants agreed that the programmes were carried out as intended.

Overall, 18 participants were invited to complete evaluation forms. There were 54 process criteria that included learning about MLAS, talking about one's stroke, asking questions and forming discussions. All participants who responded felt these were met with the exception of three criteria, where in each case only one or two participants felt otherwise. Similarly, there were 20 outcome criteria that were largely concerned with different aspects of knowledge about stroke. There were only two criteria which a responder felt were not met relating to

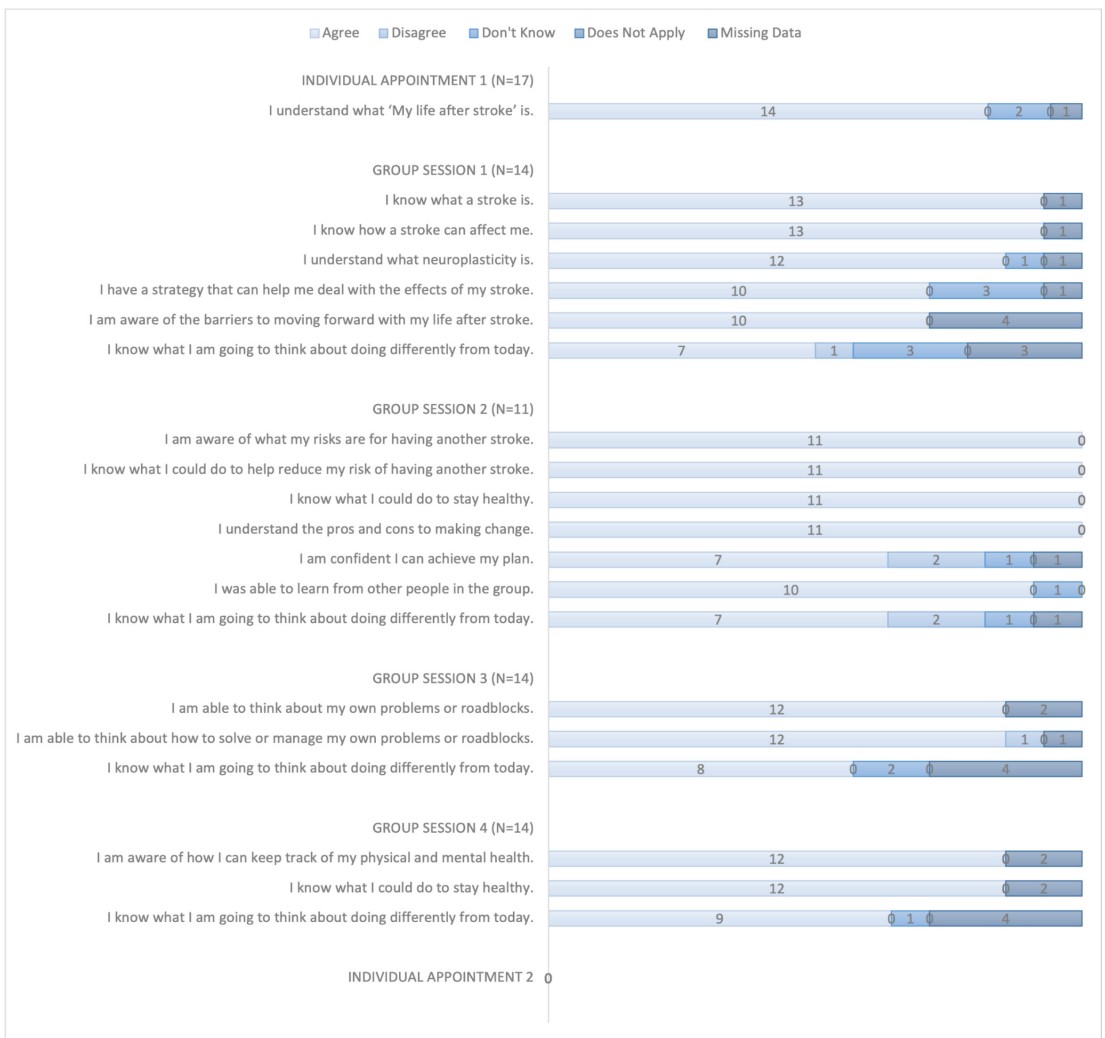

**Figure 4** Patients' evaluation forms; outcome questions. N=number of participants attending the session—individual appointment two had no outcome questions.

confidence in building action plans and acting on them. Free text collected from these forms (n=14) was positive and showed participants had 'interesting discussions', 'thoroughly enjoyed the course', found it 'useful' and enjoyed 'meeting fellow members and hearing their experiences'. One participant said 'I have really pushed myself to get past my roadblock and have started to use my quadstick to move around short distances (with somebody close by). I am sure I would not have attempted this without attending these little meetings'.

### What are participants' views on MLAS?
The interviews with participants who attended MLAS generated four themes: programme characteristics influence programme participation, programme content is relevant, perceived benefits of attendance and personal circumstances affect programme participation.

### Programme characteristics influence programme participation
Participants (n=4) spoke about the logistics of programmes, referring to preferred days of the week, time of day, location (closer to home the better) and

transport (taxis benefited participants with mobility issues). Participants (n=5) also discussed the structure of group sessions (some enjoyed groups and others did not), size of groups (small groups preferred—as delivered) and length of sessions (2 hours was long enough). One participant said 'I think six [participants] was quite nice' (female, 68, completed MLAS). Another participant said: 'I can't cope when there's lots of people in the room and too much information is coming in…too many people's voices' (female, 56, completed MLAS).

### Programme content is relevant
Despite preconceptions that the course content might not be relevant, participants, on engaging with the programme content, reported that it was. Some participants were sceptical about the activities, such as using a roadmap metaphor (ie, where participants were tasked with placing themselves on a printout of a road going in various directions) to aid thinking through barriers to recovery. One reported saying to their niece: 'what on earth are they on about being in a bleeding taxi?' But

then she went on to say: 'I didn't know what my road blocks were until I worked through the book, and the same answer kept coming up, and it was fear, fear, fear…. So once I'd said in the group, my biggest thing was fear. that was a major unblocking'. Another participant: 'Kept saying to myself: how relevant is that now?' But then said: 'It kicked my backside! I wasn't actually doing some of the things I should be doing, like …the clarinet…(and an) exercise bike' (male 90; completed MLAS). Participants (n=6) reported that programme content and delivery should meet attendees' individual needs. One participant commented, 'not everyone's difficulties are so obvious' (female 81; partially completed MLAS) as participants had their stroke at different times in their lives resulting in various after effects. Another participant explained, 'I struggle a lot with my memory…I can forget, you know, talking to you, I'd forget your name at the end' (female 58; partially completed MLAS). The contributions of individual participants (eg, identifying their own road blocks, as above) enabled content to match need. By remaining flexible in delivery, participants were able to complete handbooks at their own pace without feeling pressured: 'So when it was time to do your workbooks, I said I'll do this later, so that was never a problem. And I think I gained more from doing it after' (female, 56, completed MLAS).

### Perceived benefits of attendance
First, participants (n=6) reported a change in self-awareness, with a less negative mindset. Second, they also (n=4) perceived change in their behaviour, including increasing exercise and engaging in hobbies; one participant said, 'I was sitting there thinking there isn't anything I can do, but then I remembered I'd got this big purple exercise ball. So, it actually made me get it out, and I've got a new sewing machine…so now that's made me buzz, because now I'm back into patchwork and quilting so I think right, I'll join a patchwork and quilting group. So that was a very positive thing that came out of it… . I'm altogether just so much different…Creatively, I'm just buzzing' (female, 56, completed MLAS). Third, participants (n=10) reported that the programme increased their understanding of stroke and how it affected people: 'People shared their experiences of it, actual strokes, what they could remember, and that to me was very interesting, realising that everybody really had a different experience, but there were things in common that we were all grappling with' (female, 73, completed MLAS). Finally, participants (n=11) considered attendance to be a positive experience, gaining social support and engaging in altruism; a participant said 'I found it so helpful just listening to other people's stories and feeling able to talk about my own feelings so that was very, very helpful' (female, 78).

### Personal circumstances affect programme participation
Participants (n=7) said individual circumstances affected their participation, including having existing health

comorbidities and how far into recovery they were. One participant said 'with my eyes it is quite difficult to do all the things that I used to do…I can't use my legs and now I can't use my eyes either' (female, 56) another participant said 'it's [been] five years since I had the stroke, but most people at the group, it was a longer time since they'd had their strokes…I know this is a new project, but I would have found it helpful to have had this course much nearer to the stroke' (female, 73, completed MLAS). On the other hand, she went on to mention that there was one woman 'who'd had a stroke [recently] and I felt perhaps that was too soon for her because she was still working things through, but you know, it's probably a personal thing'.

### What feedback was given after MLAS was delivered?
Informal feedback was received from participants (n=2) and facilitators (n=10) via emails after attending an MLAS programme and grouped into 3 themes.

### Participants had a positive experience
Participants expressed positive experiences and found the programme useful. One said he was 'sceptical at first about joining the course and felt it would be a waste of time' but has 'learnt so much and is glad he attended' (70, male). Another mentioned it was 'good to speak with other stroke survivors' and found problem solving 'most useful when it came to putting it into real life' (78, female). A facilitator said that one participant didn't realise how angry he was, but by the second group session 'felt like a weight had been lifted' (Northamptonshire programme). Facilitators recognised participants were positive when completing the programmes as they had learnt from each other and shared personal experiences.

### Programme more beneficial closer to time of stroke onset
Participants' strokes were often not recent, one stated that the programme 'would be more useful when you come home from hospital'. Facilitators explained how participants struggled to identify areas of difficulty as they no longer fitted the stereotypical stroke survivor.

### Ongoing health issues
Facilitators expressed the view that many of participants had ongoing health comorbidities which were often more pronounced than the consequences of their stroke.

## DISCUSSION AND CONCLUSION
### Key findings
Participants who attended MLAS had a positive experience and felt that the programmes achieved what was intended. However, over half of those invited declined to attend. Reasons included: good recovery, ongoing health issues, logistical issues or not feeling the programme was appropriate for them.

## Strengths and limitations

More participants (n=141) took part in MLAS compared with evaluations of other self-management programmes for stroke.[22–24 26 27 34] Using a mixed methods approach helped to identify reasons for low participation, and explore why those who had attended MLAS felt it was a success. Furthermore, consistent themes were identified from the different sources of data. All qualitative data were double coded, with a third experienced analyst resolving disagreements. Only 32% of people who declined to take part in a programme gave an explanation as to why this was, but these data were supplemented by five interviews with those who did not attend. Only a small sample of attendees received an evaluation form (13%) from four programmes (17%), to minimise participants' burden. The use of emails provided a perspective from the facilitators, which complemented that of participants. However, there was a lack of diversity within the sample, participants being primarily white, retired and more than a year post stroke. Participants may not have been representative of the whole population of stroke survivors as they were all taking part in a randomised controlled trial.

## Comparison with existing literature

There is some evidence from randomised controlled trials that self-management programmes may improve quality of life and self-efficacy.[15] With regard to patients' experience and participation, there have been insights from before and after studies and randomised control trials (RCTs). The New Start intervention,[34] which has similarities to MLAS, is a self-management programme for patients within the first 6 months of their stroke which is being tested in an ongoing RCT.[22 23] The SSMP intervention[24] was part of an RCT that aimed to recruit participants of similar age to MLAS>3 months post stroke to an 8-week programme; 70% of attendees were >12 months post stroke, which is similar to MLAS. The MOST intervention[26] compared a new self-management programme with 34 hours of group contact time over 8 weeks to a standard education programme>3 months post stroke. It involved three times as much contact time as MLAS using similar sized groups and ages. MLAS does appear to have a slightly lower uptake than these programmes but a higher completion rate. The SSMP intervention[24] had 53% (27/48) attendance compared with 34% (141/420) in MLAS, with 52% (25/48) completing>50% of sessions compared with MLAS, which had 90.1% of participants complete more than 50% of sessions. This could be due to participants receiving dates in advance and having transport provided. The MOST intervention[26] had a small 60% (18/30) participants' recruitment and only 22% (4/18) attended all of the sessions compared with 53.2% for MLAS. Qualitative feedback from participants of these other programmes was positive, as it was for MLAS. Reasons for non-attendance of the SSMP course (time constraints, lack of interest and inability to attend, eg, because of transportation issues) were similar to MLAS. A follow-up study to MOST,[26] MOST-TR[27] involved video conferencing with 36 hours of online-contact time over a 9-week period with participants 3–18 months post stroke. Participants valued access to this programme without the need to travel. A review[15] of the effectiveness of self-management programmes for community living adults with stroke identified 14 RCTs conducted across 8 countries between 2000 and 2015 with 1863 participants. All programmes were community, home or outpatient based and lasted a duration of 4 weeks to 6 months. There was some evidence of improved quality of life (the primary outcome). Secondary outcome measures showed positive improvements in self-efficacy, self-esteem, emotional stability, anxiety and depression and mood.

## Interpretation

Given that there is some evidence that self-management programmes can improve quality of life in stroke survivors,[15] it is important to understand why people do or do not attend such programmes. While people who attended the MLAS programme found it helpful, many people in our sample did not feel that the programme was suitable for them. Some people felt that it had been too long since their stroke had passed to be useful to attend, though other programmes that recruited participants earlier after their stroke also had poor uptake.[24 26 27] It is a feature of primary care that most people on stroke registers will not have had a recent stroke, and so this raises the question as to whether self-management programmes might be more useful if accessed from specialist secondary care services or offered at the point of discharge or engagement with early supported discharge. Other potential participants felt they had too many ongoing health problems to attend, highlighting the problem of providing single-condition programmes for people with multimorbidity.[35 36] For those who do wish to attend a programme, it needs to be adaptable to patient needs and wishes. For example, it has been shown to be feasible to provide virtual sessions which can overcome attendance barriers[37]. Tailoring programmes might also address concerns of people who do not like group work. It is possible the decision to take part in an MLAS programme was influenced by how it was introduced to the stroke survivor by the healthcare professional. It may be that uptake could be improved by focussing on this aspect. Before widespread adoption of self-management programmes such as MLAS, further evidence is required on optimal timing, content and mode of delivery.[15]

## Conclusion

The mixed methods and multiperspective nature of this analysis has provided insights into patients' experience of and participation in a stroke self-management programme, MLAS. This intervention, which focuses on addressing patients' self-reported needs, was found to provide a positive experience for participants. Although many stroke survivors who declined participation did not feel it was appropriate for them, it might have a role as part of a menu of optional services.

## Practical implications

This study suggests that there should not be an expectation that all stroke survivors would want to take part in a self-management programme such as MLAS. There was feedback that the programme would be more valuable if offered sooner after the stroke. This could be achieved by having referring routes from specialist services rather than from primary care, though uptake is not necessarily higher from these settings. Some potential participants did not like the idea of group work, and others found it logistically difficult to attend. Setting up online sessions (whether individual or group based) would be a way of addressing such concerns.

**Acknowledgements** We are thankful for the Improving Primary Care After Stroke administration team members who included Lizzie Kreit, Victoria Theobald and Alex Newbrook, and to James Brimicombe for his database work. With thanks to the trainers from Leicester and the my life after stroke facilitators, who without them, the programmes would not have been delivered in the community venues across England. Thank you to the stroke survivors participating in the study. Thank you to Bundy Mackintosh, patients' adviser.

**Contributors** EGB: conceptualisation, methodology, investigation, data curation, writing—original draft and visualisation. MRJA: conceptualisation, methodology, writing—review & editing, project administration and supervision. JG: investigation, data curation and writing—review & editing. VJ: conceptualisation, resources and writing—review & editing. RM: conceptualisation, writing—review & editing, supervision, project administration and funding acquisition. LL: conceptualisation and writing—review & editing. JM: Guarantor, conceptualisation, writing—review & editing, supervision and funding acquisition.

**Funding** This study is funded by the National Institute for Health Research's Programme Grant for Applied Research titled 'Developing primary care services for stroke survivors' reference PTC-RP-PG-0213-20001. The views expressed are those of the authors and not necessarily those of the NIHR or the Department of Health and Social Care. JM is an NIHR senior investigator.

**Competing interests** VJ declares that she is employed by the University Hospitals of Leicester NHS Trust within the Leicester Diabetes Centre (LDC), which receives not-for-profit income for DESMOND, a suite of self-management programmes for which LDC holds the Intellectual Property Rights for. UHL also receives various grants to pay staff within LDC to carry out and implement various studies and self-management programmes. The Intellectual Property for MLAS is held by University of Leicester on behalf of UHL.

**Patient and public involvement** Patients and/or the public were involved in the design, or conduct, or reporting, or dissemination plans of this research. Refer to the Methods section for further details.

**Patient consent for publication** Not applicable.

**Ethics approval** This study involves human participants. It does involve human participants and has been approved by an Ethics Committee, Yorkshire & The Humber-Bradford Leeds NHS Research Ethics Committee (17/YH/0441). Participants gave informed consent to participate in the study before taking part.

**Provenance and peer review** Not commissioned; externally peer reviewed.

**Data availability statement** Data are available upon reasonable request. data available upon reasonable request.

**ORCID iD**
Emily Grace Blatchford http://orcid.org/0000-0003-3429-402X

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
