## [Reviewer comments · BMJ Open]

ARTICLE DETAILS

TITLE (PROVISIONAL)	Patient experience of and participation in a stroke self-management programme, My Life After Stroke (MLAS): a multi-method study
AUTHORS	Blatchford, Emily; Aquino, Maria; Grant, Julie; Johnson, Vicki; Mullis, Ricky; Lim, Lisa; Mant, Jonathan

VERSION 1 – REVIEW

REVIEWER	Crocker, Thomas Bradford Teaching Hospitals NHS Foundation Trust, Academic Unit for Ageing and Stroke Research
REVIEW RETURNED	18-Apr-2022

GENERAL COMMENTS	The authors have reported an informative and useful evaluation of participant experience of the My Life After Stroke intervention. The intervention is an important development, attempting to address a guideline recommendation for which there is little evidence of effective interventions. The findings are generally well-presented - I particularly like the illustration of the themes - and appropriately discussed. I have suggested minor revisions. Substantive comments: Methods: p6, l32-34. Can you provide the topic guide? It would be useful to understand what the interviews explored, and how this may have shaped the findings. Results: p8, l44 and p9, l17. Five interviews are referred to, and later 12 with a broader inclusion criteria that may have encompassed the earlier five. Please clarify if these are referring to some of the same interviews or if they were separate. Either way, please mention the interviews in Why did participants decline MLAS? in the Methods. p9, l17. Can you break down the numbers for the categories provided (invited but did not attend, attended but did not complete, completed) and, when reporting the themes, relate these to the participants who were relevant to each (presumably those who did not attend did not contribute to some themes)? p9, l34-43. Was any more said about programme content? Particular bits they especially liked or considered helpful or thought would have been better without? This section is positive overall but seems to lack detail given the objective of the paper. Discussion: Strengths and limitations. Please mention aphasia and cognitive impairment and how inclusion of people with these conditions was facilitated or otherwise.
--

	Additional comments: p4, l12-14. Is it possible to provide some more up-to-date references in relation to psychological issues and their consequences? p6, l8. There seem to be some missing words such as "and so" in "...attend the programme [] were unable..." p6, l23. I found the evaluation form (pp25-30 of the PDF) but it didn't seem to be labelled Appendix B and there's an appendix B in the protocol. p7, l24. It would be nice to get a sense of the average number of participants in a group and perhaps average attendance for each session as it relates to the 'smaller is better' perception. p7, l27-28. The percentages given don't seem to match the phrasing of the sentence. Please clarify what the percentages refer to. p7, l59. Presumably you didn't collect time since stroke for non-attendeess, which is unfortunate since you want to claim time since stroke was relevant. p9, l19-20: Programme Characteristics Influence Participation, and, Personal Circumstances Affect Participation. Participation in this context is ambiguous (could refer to life/social/functional participation). Consider 'programme participation'. p9, l28-29. Can you clarify smaller than what? e.g. they would have preferred smaller than how it was delivered or were glad it was as small as it was or both? p9, l34-43: Programme Content Is Important. Appropriate, relevant or both seem like better descriptors than important. Also, process as well as content appear to be referred to in this theme (e.g. flexibility). p9, l36-37. The sentence is ambiguous. Presumably they thought this programme would/did meet needs not just should with different content for example? Were participants referring to their needs, projecting about others' needs, or both? p10, l58-60. Strictly, you don't know over half declined for these reasons (64 of 228 gave reasons). Consider using "; reasons included..." Note: I have generally used the page numbers provided in the author manuscript, which are one less than those over-printed in the PDF.
--	---

REVIEWER	Geraghty, Adam
REVIEW RETURNED	University of Southampton, Primary Care and Population Sciences 26-May-2022

GENERAL COMMENTS	Thank you for the opportunity to review this paper. Generally, the report is well written and covers the topic well. Thus, my review will be brief. I have a few points to consider/address: Methodology: In the analysis on page 6, the authors write that they conducted an inductive thematic analysis. First, it would be good to clarify what particular approach they used with thematic analysis. There are many different approaches to thematic analysis, and the references provided don't link to a consistent approach? E.g. some are Braun and Clarke papers, others are not? I know Braun and Clarke now have a really quite specific approach to thematic analysis, and I don't think that has been used here. Please could the authors clarify?
--

	Additionally, the line: “Two coders (EB, JG) individually coded the interview data for phrases related to MLAS, including terms of enjoyment, learning etc., which were summarised and categories were formed” Doesn't sound that inductive to me? E.g. coding for specific terms? Could that be expanded and explained? Often with inductive coding one will code everything, without looking for specific phrases or aspects. This may come much later, but not initially. Analysis: Overall, the analysis is useful. However, some of the themes currently felt a bit 'thin'? I realise this is not an interpretive analysis. Still, this is a complex, interesting area. As a reader I wondered if there was more to say or unpack/expect on the following theme: “Programme Content Is important Participants (n=6) reported that programme content should meet attendees' obvious” [female, 81] as participants had their stroke at different times in their lives resulting in various after-effects. Another participant explained, “I struggle a lot with my memory...I can forget, you know, talking to you, I'd forget your name at the end” [female, 58]. By remaining flexible, participants were able to complete handbooks at their own pace without feeling pressured” Any more on the content? Any additional quotes? Other aspects that came up? There may not be. But if possible, I think it would add impact if this very important section (for others who are developing similar interventions) were expanded and developed further.
--	---

VERSION 1 – AUTHOR RESPONSE

Reviewer 1

p6, l32-34. Can you provide the topic guide? It would be useful to understand what the interviews explored, and how this may have shaped the findings. - Now provided as Appendix C.

p8, l44 and p9, l17. Five interviews are referred to, and later 12 with a broader inclusion criteria that may have encompassed the earlier five. Please clarify if these are referring to some of the same interviews or if they were separate. Either way, please mention the interviews in Why did participants decline MLAS? in the Methods.

Sorry – it is confusing as written. The 12 and the 5 are separate patients. We have added the following sentence to the methods to clarify:

- The qualitative data included a sub-set of interviews that had been carried out for the wider process evaluation[31]. This included 8 participants who completed MLAS, 4 who started, but did not complete, and 5 who did not take part (two of whom had expressed interest).

And under the heading 'why did participants decline MLAS?', we have edited the following sentence:

- In addition to interviews with 5 non-participants, Participants were sent a questionnaire was sent to all participants 6-months ...

This clarification has made the following text redundant in the heading 'what are participant views on MLAS?':

-Potential participants were selected to ensure that people who completed a programme, started a programme but did not complete it, declined to take part, or agreed to take part but did not attend were all included.

It has also enabled us to edit the results sections which now read:

- The Interviews were conducted with participants who had been invited to MLAS but declined (n=3) or expressed interest but did not attend (n=2). MLAS gave Reasons matching those in Figure 2, with additional ones including

- The Interviews were conducted with a small sample of participants who had attended and/or completed MLAS (n= 12) Four themes were identified four themes:

p9, l17. Can you break down the numbers for the categories provided (invited but did not attend, attended but did not complete, completed) and, when reporting the themes, relate these to the participants who were relevant to each (presumably those who did not attend did not contribute to some themes)?

Numbers now provided, as per response to previous comment. The 4 non-participants contribute to the section on why participants declined MLAS, the 11 participants who took part in MLAS contribute to the section on participant views on MLAS. Where we have used a quote, we have added whether or not this was from a completer or partial completer.

p9, l34-43. Was any more said about programme content? Particular bits they especially liked or considered helpful or thought would have been better without? This section is positive overall but seems to lack detail given the objective of the paper.

We have added detail to this paragraph. The whole paragraph now reads:

Despite pre-conceptions that the course content might not be relevant, participants, on engaging with the programme content, reported that it was. Some participants were sceptical about the activities, such as using a roadmap metaphor (i.e where participants were tasked with placing themselves on a printout of a road going in various directions) to aid thinking through barriers to recovery. One reported saying to their niece: "what on earth are they on about being in a bleeding taxi?" But then she went on to say: "I didn't know what my road-blocks were until I worked through the book, and the same answer kept coming up, and it was fear, fear, fear....So once I'd said in the group, my biggest thing was fear.. that was a major unblocking". Another participant: "Kept saying to myself: how relevant is that now?" But then said: "It kicked my backside! I wasn't actually doing some of the things I should be doing, like ..the clarinet..(and an) exercise bike" (male 90; completed MLAS). Participants (n=6) reported that programme content and delivery should meet attendees' individual needs. One participant explained commented, "not everyone's difficulties are so obvious" [female 81; partially completed MLAS] as participants had their stroke at different times in their lives resulting in various after-effects. Another participant explained, "I struggle a lot with my memory...I can forget, you know, talking to you, I'd forget your name at the end" [female 58; partially completed MLAS]. The contributions of individual participants (e.g. identifying their own road-blocks, as above) enabled content to match need. By remaining flexible in delivery, participants were able to complete handbooks at their own pace without feeling pressured: "So when it was time to do your work books, I said I'll do this later, so that was never a problem. And I think I gained more from doing it after" [female, 56, completed MLAS].

Strengths and limitations. Please mention aphasia and cognitive impairment and how inclusion of people with these conditions was facilitated or otherwise.

We have added the following to the methods to explain our approach to this issue:

-In order to facilitate engagement of people with aphasia or cognitive impairment, participants could select how they were interviewed (with or without carer; by telephone or face to face or by email).

Additional comments:

p4, l12-14. Is it possible to provide some more up-to-date references in relation to psychological issues and their consequences?

We have replaced the older two references with:

- Allida S, Cox KL, Hsieh C-F, House A, Hackett ML. (2020) Pharmacological, psychological and non-invasive brain stimulation interventions for preventing depression after stroke.
- Towfighi A, Ovbiagele B, Hussein NE et al on behalf of the American Heart Association Stroke Council; Council on Cardiovascular and Stroke Nursing; and Council on Quality of Care and Outcomes Research. (2017) Post-stroke depression: a scientific statement for healthcare professionals from the American Heart Association/American Stroke Association. Stroke 48:e30-e43.

p6, l8. There seem to be some missing words such as "and so" in "...attend the programme [] were unable..."

We have amended the sentence which now reads:

- In some areas, there were delays with set-up of MLAS which meant that participants' received the questionnaire before their invitation to attend the programme and so were unable to provide a response to this question.

p6, l23. I found the evaluation form (pp25-30 of the PDF) but it didn't seem to be labelled Appendix B and there's an appendix B in the protocol.

We have added a heading to this evaluation form so that it is now labelled Appendix B. The same issue applies to appendix A, so we have also added a heading to this form.

p7, l24. It would be nice to get a sense of the average number of participants in a group and perhaps average attendance for each session as it relates to the 'smaller is better' perception.

We have added the following text to describe this:

- Twenty four separate MLAS programmes were held, The median attendance at a group session was 4.5 (range 2-8).

p7, l27-28. The percentages given don't seem to match the phrasing of the sentence. Please clarify what the percentages refer to.

We have corrected this to:

- Of those expressing interest, 141 (3473%) attended, and 103 of those attending (73%) completed the programme.

p7, l59. Presumably you didn't collect time since stroke for non-attendees, which is unfortunate since you want to claim time since stroke was relevant.

We have amended table 1 to represent time since stroke to study entry (rather than time since stroke to attending MLAS), so that we can report this for non-attendees aswell. We have also tested for significance. Text amended:

- The mean distance from the programme venue to the stroke survivor's home was and time since stroke were not significantly different between those who did and did not attend the programme (p-value 0.26 and 0.50 respectively).

For completeness, we have added standard deviations to the table.

p9, l19-20: Programme Characteristics Influence Participation, and, Personal Circumstances Affect Participation. Participation in this context is ambiguous (could refer to life/social/functional participation). Consider 'programme participation'.

We have amended as suggested:

- Programme characteristics influence programme participation, Personal circumstances affect programme participation

p9, l28-29. Can you clarify smaller than what? e.g. they would have preferred smaller than how it was delivered or were glad it was as small as it was or both?

Clarification provided (they liked the small sizes of the groups that were provided):

- size of groups (smaller groups preferred – as delivered)

p9, l34-43: Programme Content Is Important. Appropriate, relevant or both seem like better descriptors than important. Also, process as well as content appear to be referred to in this theme (e.g. flexibility).

We have revisited what we meant by the phrase 'programme content is important'. The key point was that participants felt the content was important despite pre-conceptions. that the flexibility enabled both participants' needs to be met, and participants to put the content into practice. We have relabelled the theme subheading to:

-Programme Content Is Important Relevant

and we have added a sentence to clarify what we mean to the beginning of the paragraph:

-Despite pre-conceptions that the course content might not be relevant, participants on engaging with the programme did concede that it was.

With regard to flexibility, this is relevant within this theme as it was a mechanism whereby the programme could meet participants' needs (and therefore be deemed to be relevant). This relates to the next reviewer comment:

p9, l36-37. The sentence is ambiguous. Presumably they thought this programme would/did meet needs not just should with different content for example? Were participants referring to their needs, projecting about others' needs, or both?

We have reworded this sentence:

- Participants (n=6) reported that programme content and delivery should meet attendees' individual needs.

We have also expanded the explanatory sentence:

- The contributions of individual participants (e.g. identifying their own road-blocks, as above) enabled content to match need. By remaining flexible in delivery, participants were able to complete handbooks at their own pace without feeling pressured: "So when it was time to do your work books, I said I'll do this later, so that was never a problem. And I think I gained more from doing it after" [female, 56, completed MLAS].

We have expanded and clarified the whole paragraph in response to reviewer 2 (see below).

p10, l58-60. Strictly, you don't know over half declined for these reasons (64 of 228 gave reasons). Consider using "; reasons included..."

We have amended this sentence as suggested so it now reads:

Reasons included: due to good recovery, ongoing health issues, logistical issues, or not feeling the programme was appropriate for them.

Reviewer 2

In the analysis on page 6, the authors write that they conducted an inductive thematic analysis. First, it would be good to clarify what particular approach they used with thematic analysis. There are many

different approaches to thematic analysis, and the references provided don't link to a consistent approach? E.g. some are Braun and Clarke papers, others are not? I know Braun and Clarke now have a really quite specific approach to thematic analysis, and I don't think that has been used here. Please could the authors clarify? -We have simplified the referencing and rewritten this section which now reads:

-We applied the six-phase approach of Braun and Clarke[33]to the data as follows: 1) data familiarisation where transcripts and questionnaires with qualitative data were read and reread; 2) generation of initial codes, where two coders independently generated codes on a subset of transcripts, then refined these and applied the codes to the rest of the data; 3) identified themes, i.e. searching for patterns within the coded data; 4) reviewing the themes to refine each theme's meaning, and to determine whether some themes lacked coherence/depth and needed discarding, 5) defining and naming themes which involved finalising the set of themes, discussing any disagreements between coders with a third member of the research team (RA); and finally 6) writing up the report which was led by the first author (EB)

Additionally, the line:

"Two coders (EB, JG) individually coded the interview data for phrases related to MLAS, including terms of enjoyment, learning etc., which were summarised and categories were formed"

Doesn't sound that inductive to me? E.g. coding for specific terms? Could that be expanded and explained? Often with inductive coding one will code everything, without looking for specific phrases or aspects. This may come much later, but not initially.

See response to previous comment.

However, some of the themes currently felt a bit 'thin'? I realise this is not an interpretive analysis. Still, this is a complex, interesting area. As a reader I wondered if there was more to say or unpack/expect on the following theme:

"Programme Content Is important

Participants (n=6) reported that programme content should meet attendees' obvious" [female, 81] as participants had their stroke at different times in their lives resulting in various after-effects. Another participant explained, "I struggle a lot with my memory...I can forget, you know, talking to you, I'd forget your name at the end" [female, 58]. By remaining flexible, participants were able to complete handbooks at their own pace without feeling pressured"

We have expanded on this theme. The paragraph has been re-labelled (see response to reviewer 1) and expanded:

-Programme Content Is Important Relevant

Despite pre-conceptions that the course content might not be relevant, participants, on engaging with the programme content, reported that it was. Some participants were sceptical about the activities, such as using a roadmap metaphor (i.e where participants were tasked with placing themselves on a printout of a road going in various directions) to aid thinking through barriers to recovery. One reported saying to their niece: "what on earth are they on about being in a bleeding taxi?" But then she went on to say: "I didn't know what my road-blocks were until I worked through the book, and the same answer kept coming up, and it was fear, fear, fear....So once I'd said in the group, my biggest thing was fear.. that was a major unblocking". Another participant: "Kept saying to myself: how relevant is that now?" But then said: "It kicked my backside! I wasn't actually doing some of the things I should be doing, like ..the clarinet..(and an) exercise bike" (male 90; completed MLAS). Participants (n=6) reported that programme content and delivery should meet attendees' individual needs. One participant explained commented, "not everyone's difficulties are so obvious" [female 81; partially completed MLAS] as participants had their stroke at different times in their lives resulting in various after-effects. Another participant explained, "I struggle a lot with my memory...I can forget, you know,

talking to you, I'd forget your name at the end" [female 58; partially completed MLAS]. The contributions of individual participants (e.g. identifying their own road-blocks, as above) enabled content to match need. By remaining flexible in delivery, participants were able to complete handbooks at their own pace without feeling pressured: "So when it was time to do your work books, I said I'll do this later, so that was never a problem. And I think I gained more from doing it after" [female, 56, completed MLAS].

Any more on the content? Any additional quotes? Other aspects that came up? There may not be. But if possible, I think it would add impact if this very important section (for others who are developing similar interventions) were expanded and developed further.

We have also slightly expanded the theme of 'perceived benefits of attendance':

So that was a very positive thing that came out of it... I'm altogether just so much different...Creatively, I'm just buzzing". Thirdly, participants (n=10) reported that the programme increased their understanding of stroke and how it affected people: "People shared their experiences of it, actual strokes, what they could remember, and that to me was very interesting, realising that everybody really had had a different experience, but there were things in common that we were all grappling with".